African penguins follow the gaze direction of conspecifics

http://orcid.org/0000-0003-4582-4057 Nawroth Christian 1 2 nawroth.christian@gmail.com
Trincas Egle 3 4
http://orcid.org/0000-0002-8698-472X Favaro Livio 3
1 School of Biological and Chemical Sciences, Queen Mary University of London , London , UK
2 Institute of Behavioural Physiology, Leibniz Institute for Farm Animal Biology , Dummerstorf , Germany
3 Department of Life Sciences and Systems Biology, University of Turin , Turin , Italy
4 Zoom Torino , Turin , Italy
Vonk Jennifer
Electronic publication date: 2017 Jun 12
Publication date: 2017
Volume: 5
Electronic Location ID: e3459
Received 2017 Feb 16; Accepted 2017 May 24
Copyright: © 2017 Nawroth et al.
Copyright year: 2017
Copyright holder: Nawroth et al.
License: This is an open access article distributed under the terms of the Creative Commons Attribution License, which permits unrestricted use, distribution, reproduction and adaptation in any medium and for any purpose provided that it is properly attributed. For attribution, the original author(s), title, publication source (PeerJ) and either DOI or URL of the article must be cited.
License URL: https://creativecommons.org/licenses/by/4.0/

Keywords: Predation, Gaze following, Spheniscidae, Social cognition

Funding: British Academy/Leverhulme Small Research Grant SG160975 Deutsche Forschungsgemeinschaft NA 1233/1-1 This work was supported by a British Academy/Leverhulme Small Research Grant to C.N. and L.F. (SG160975) and by a fellowship from the Deutsche Forschungsgemeinschaft (NA 1233/1-1) to C.N. and L.F. was also supported by the University of Torino through a System S.p.A. research grant for bioacoustics. The funders had no role in study design, data collection and analysis, decision to publish, or preparation of the manuscript.

==============================
Gaze following is widespread among animals. However, the corresponding ultimate functions may vary substantially. Thus, it is important to study previously understudied (or less studied) species to develop a better understanding of the ecological contexts that foster certain cognitive traits. Penguins (Family Spheniscidae), despite their wide interspecies ecological variation, have previously not been considered for cross-species comparisons. Penguin behaviour and communication have been investigated over the last decades, but less is known on how groups are structured, social hierarchies are established, and coordination for hunting and predator avoidance may occur. In this article, we investigated how African penguins (Spheniscus demersus) respond to gaze cues of conspecifics using a naturalistic setup in a zoo environment. Our results provide evidence that members of the family Spheniscidae follow gaze of conspecifics into distant space. However, further tests are necessary to examine if the observed behaviour serves solely one specific function (e.g. predator detection) or is displayed in a broader context (e.g. eavesdropping on relevant stimuli in the environment). In addition, our findings can serve as a starting point for future cross-species comparisons with other members of the penguin family, to further explore the role of aerial predation and social structure on gaze following in social species. Overall, we also suggest that zoo-housed animals represent an ideal opportunity to extend species range and to test phylogenetic families that have not been in the focus of animal cognitive research.

Introduction

Gaze following is widespread in the animal kingdom and has been reported for a variety of nonhuman primates (Rosati & Hare, 2009), but also other mammals (Kaminski et al., 2005; Werhahn et al., 2016), birds (Schloegl, Kotrschal & Bugnyar, 2007), and reptiles (Wilkinson et al., 2010). However, although several species share this skill, the corresponding ultimate functions may vary and at least three different cognitive mechanisms have been classified: gaze following into distant space, geometrical gaze following and shared attention (Emery, 2000; Davidson et al., 2014). The first is defined as the co-orientation of one’s own gaze direction into distant space with that of another individual (Tomasello, Call & Hare, 1998; Wilkinson et al., 2010). This form of visual co-orientation is an automatic, reflexive shift of gaze in response to another individual gazing to search for anything interesting along this line of sight (Gómez, 2005). On the other hand, geometric gaze following enables a subject to follow the gaze of another individual around barriers—an ability that would require a subject to take the visual perspective of another individual (Bräuer, Call & Tomasello, 2005). A third form of following the gaze direction of another individual is to share attention towards an external focal object or event (Gómez, 2005).

Investigations on gaze cues have been a major focus in primate research (Rosati & Hare, 2009). For instance, great apes and several monkey species know what conspecifics or humans can see (Hare et al., 2000; Bulloch, Boysen & Furlong, 2008; Botting, Wiper & Anderson, 2011; Overduin-de Vries, Spruijt & Sterck, 2013). This is not surprising, given that animals that live in groups would benefit from cognitive skills that allow them to monitor and eavesdrop on other individuals, and would thus be able to compete better with others over critical resources (Kummer et al., 1997). Recently, social structure and group size in primates have been linked to the interpretation of gaze cues (Tomasello, Call & Hare, 1998; MacLean et al., 2013). For example, species of lemurs living in larger groups followed the gaze of an experimenter into distant space more often than lemurs from species that have less complex social systems (Sandel, MacLean & Hare, 2011).

In birds, most work on gaze cues and gaze following has been done on members of the corvid family (Seed, Emery & Clayton, 2009; von Bayern & Emery, 2009). It has been found that ravens (Corvus corax) follow the gaze of conspecifics into distant space (Bugnyar, Stöwe & Heinrich, 2004; Schloegl, Kotrschal & Bugnyar, 2007), but are also able to follow gaze around barriers (Bugnyar, Stöwe & Heinrich, 2004). Recently, other non-corvid birds such as Northern bald ibises, Geronticus eremita (Loretto, Schloegl & Bugnyar, 2010), Greylag geese, Anser anser (Kehmeier et al., 2011), and European starlings, Sturnus vulgaris (Butler & Fernández-Juricic, 2014) have also been found to follow gaze of conspecifics. Gaze following in altricial birds, such as corvids, seems to emerge shortly after fledging (Schloegl, Kotrschal & Bugnyar, 2007)—a time when individuals experience higher exposure to aerial predation. The so-called ‘predation hypothesis’ states that gaze following into distant space, at least in birds, might be a protective measure against predators. In turn, we would expect that species exposed to high levels of aerial predation should be more prone to follow the gaze of others into distant space than subjects with lower predation risks (Gómez, 2005). However, there are no studies available that directly link risk of predation and individual’s inclination to follow gaze. Thus, more genera and species are required in comparative cognition research to develop a better understanding of the ecological contexts that foster certain cognitive traits and to track their evolutionary origins (Shettleworth, 2009).

Penguins (Aves, Sphenisciformes, Spheniscidae) are a family of seabirds that have evolved to swim and have lost the ability to fly (Williams, 1995). The common ancestry of penguins dates back to about 40 mya and the family is now divided into six extant genera (Baker et al., 2006). Over the last decades, penguin behaviour gained attention, particularly in terms of their abilities to differentiate conspecifics using their vocalisations for relocating mates and offsprings in densely packed groups (Aubin, Jouventin & Hildebrand, 2000; Clark, Boersma & Olmsted, 2006). However, less is known on how groups are structured and social hierarchies are established (but see Ancel et al., 2015). There are also no studies available that investigate the use of gaze cues in social contexts (e.g. during foraging or predator detection/avoidance). In this article, we tested African penguins for their ability to respond to gaze cues of conspecifics in a semi-naturalistic setting. African penguins are social animals, including a complex system of vocal communication (Favaro, Ozella & Pessani, 2014; Favaro et al., 2015), and are exposed to aquatic, terrestrial and aerial predators (Pichegru, 2013)—both factors that are considered to affect the interpretation of gaze cues from conspecifics (Gómez, 2005; Sandel, MacLean & Hare, 2011; MacLean et al., 2013). Given the exposure of the African penguin to such ecological sources of selection, we expected this species to be able to co-orient with conspecifics’ gaze.

Methods

Ethical note

The research was carried out with permission from Zoom Torino S.p.A. (https://www.zoomtorino.it). This zoological institution has high standards for animal welfare and is accredited by the EAZA (European Association of Zoos and Aquaria) and UIZA (Unione Italiana Giardini Zoologici e Acquari). Because all recording procedures were non-invasive and did not cause any disturbance to the animals, the approval of an ethics committee was not required by Italian laws. Animal care and all experimental procedures were in accordance with the ASAB/ABS Guidelines for the Use of Animals in Research (Association for the Study of Animal Behaviour, 2016).

Subjects and housing

A total of 45 African penguins were tested (21 males, 24 females), aged 1–29 years (mean ± SEM: 8.2 ± 1.1). The birds were all members of a large ex situ colony (i.e. 64 individuals on October 2016) hosted in a semi-naturalistic enclosure (1,500 m2 and a swimming pool of 120 m2) at the biopark Zoom Torino, Cumiana (TO), Italy (44°56′N, 7°25′E). Daily care for the animals was provided by zoo employees and volunteers.

Experimental procedure

All the tests were carried out in September and October 2016 (i.e. a period where the impact of the viewing public is minimal to the study group; Ozella et al., 2015). The experiment was conducted by a single human (Egle Trincas) who was equipped with a laser pointer and was standing outside the penguin enclosure (similar to an experiment on Northern bald ibises: Loretto, Schloegl & Bugnyar, 2010). In test trials (T), the experimenter waited for the occurrence of spatial configurations in which one penguin (the test subject) was facing towards the experimenter, and another subject (the demonstrator) was in a position where it is in front of the test subject with its back turned towards the experimenter (Fig. 1). To control for stimulus and local enhancement, the spatial configuration had to be as follows: demonstrator—test subject—laser dot (Fig. 1). The experimenter projected for a maximum of 3 s (or as soon as the demonstrator looked towards) a laser dot onto a close obstacle (e.g. a wall or a wooden fence), and scored the behaviour of the test subject for the following 5 s. Because penguins show vigilance behaviour while resting (i.e. observing/screening their environment), we administered a first control condition (C1) to exclude accidental gaze co-orienting between two subjects. The initial constellation of the two subjects (demonstrator and test subject) was identical as in test trials, but no laser dot was presented. An additional second control condition (C2) was administered to exclude the possibility that the tested subjects changed behaviour because they responded to the laser dot instead of the change in gaze direction of the demonstrator. In this control condition, no demonstrator was present, and the experimenter projected for 3 s a laser dot onto a close obstacle (e.g. wall or wooden fence) behind the test subject, and scored its behaviour for the following 5 s. The distance between demonstrator and test subject in T and C1 trials varied (approximately 5–20 m), as several potential spatial locations in the enclosure were used. This was also the case for the distance between the experimenter and the demonstrator or test subject, which, depending on the position of the animals in the enclosure, varied as well (approximately 0.5–4 m). Some subjects were tested more than once due to the opportunistic nature of the test design and received trials of each condition in a random and opportunistic order. This resulted in the fact that demonstrator and test subject were not necessarily the same in T and C1 trials. However, subjects received a maximum of one trial an hour. Each bird had flipper bands for individual recognition and the experimenter reliably identified individuals because of previous experience in handling them.

Figure 1 Experimental setup for test (T) and control (C1) trials.

The ‘test subject’ had to be oriented towards the experimenter, while the demonstrator had to be positioned between experimenter and test subject. Additional control trials (C2) looked identical, but without a demonstrator subject.

Data coding and statistical analysis

All trials were videotaped from the experimenter’s position using a Sony HDR-SR5E HD camcorder. Subjects’ responses in trials were scored through video analysis using BORIS (Behavioral Observation Research Interactive Software (Friard & Gamba, 2016)). For the test condition and C2, we analysed whether the test subject changed gaze and was looking in the same direction where the laser dot was presented, within a maximum of 8 s (3 s dot presentation + 5 s additional response time without dot), after a trial started. For C1, we chose a 10 s sequence of the test subject and the demonstrator, and screened the entire sequence for gaze-orienting behaviour by the focal test subject. As T and C1 trials needed the same spatial configuration of demonstrator and test subject, the experimenter randomly assigned specific situations to either T or C1 trials. For T trial analysis, we also included all trials in which the demonstrator did not look towards the laser dot after presentation, because we could not exclude that the demonstrator might have expressed more subtle head orientation cues that could not have been easily spotted by the experimenter. We used beak direction of test subjects to assess the direction in which penguins gazed. Test subjects had to move their head at least 45° and the duration of movement had to be below 1 s. Using this approach, we could avoid including vigilance behaviour, which usually involves slower movements of the head. Penguins hunt fish and marine invertebrates (Williams, 1995), so we expected them to use binocular vision when they had to focus on specific objects or events (Martin, 2009). Thus, we assume that beak orientation is a reliable measure of attention for this species. A substantial percentage of the trials (>40%; 57/129 trials) was randomly chosen and was then coded by a second observer. Inter-observer reliability for gaze-orienting behaviour was very good (Cohen’s κ = 0.879). A generalised linear mixed model (GLMM) with binomial logit link, using the glmer function of the R package lme4, was used for analysis. Test subject response (i.e. looking towards the target location) was binary scored with either ‘yes’ or ‘no’ and was used as response variable. Because we opportunistically had to choose potential pair settings, total trial number of subjects ranged between 1 and 7 trials (mean ± SEM: 2.867 ± 0.273), leading to a total of 49 T trials, 46 C1 trials and 33 C2 trials. Test condition, age and sex of the tested subjects were included as fixed factors, while identity of tested subjects and demonstrator subjects was included as random factor, thus taking into account repeated measures for individual subjects. Before running the models, we excluded the occurrence of collinearity among predictors by examining the variance inflation factors (vif package; Fox & Weisberg, 2011). All the predictors showed vif < 2.

Results

We analysed a total of 49 test trials (T) and 46 control trials (C1). In T trials, the demonstrator subjects looked towards the laser dot in 44/49 trials (89.8% of all test trials). We found a significant effect of condition (GLMM: χ2 = 11.658; df = 1; P < 0.001) on penguins’ response rate between T and C1 trials (Fig. 2). Subjects co-oriented with the gaze direction of a conspecific significantly more often in the test than in the control condition (T vs. C1: 24.5% vs. 2.1% of the trials; see Video S1). We found no effect of sex (GLMM: χ2 = 2.146; df = 1; P = 0.143) or age (GLMM: χ2 = 2.284; df = 1; P = 0.131) on gazing behaviour. Additional control trials (C2; N = 33) were conducted to exclude that test subjects’ behavioural change was due to following the gaze direction of conspecifics and not because they perceived the laser dot behind them in T trials. In none of the C2 trials did the birds turn their gaze towards the laser dot (0% of trials). Because none of the subjects responded in C2 trials, we did not include these trials in the GLMM model.

Figure 2 Relative response rate of test subjects for the three different test conditions (Tsubjects = 33, C1subjects = 29, C2subjects = 18; Ttrials = 49, C1trials = 46, C2trials = 33).

Bars represent mean ± SE.

Discussion

We investigated the ability of penguins to follow the gaze direction of conspecifics using a semi-naturalistic setup in a zoo environment. Our results provide evidence that African penguins co-orient with the gaze of other penguins, while we ruled out that inadvertent cueing and spontaneous co-orienting account for our results.

Penguins in our experiment followed the gaze of conspecifics in 24.5% of the test trials. This is a striking difference to the relative absence of the same behaviour in our control conditions, but can be considered a rather small rate when compared with the response rate in other bird species. For example, young ravens showed a positive response in 40% of test trials (Schloegl, Kotrschal & Bugnyar, 2007), while positive response rate for Northern bald ibises was above 60% (Loretto, Schloegl & Bugnyar, 2010). One explanation for these differences might be a rather low inclination of African penguins to follow gaze of conspecifics. Alternatively, environmental cues might have distracted test subjects. Subjects were tested in their home enclosure, exposing them to potential visual, but also acoustic, cues from a large number of conspecifics. Environmental noise and movement might have further diverted subjects’ attention. Finally, penguins have laterally positioned eyes, so we cannot exclude the possibility that penguins might perceive the head movement of the demonstrator without necessarily co-orienting their head in the same line. However, penguins hunt fish and marine invertebrates (Williams, 1995), so we would expect them to use binocular vision when they have to focus on specific objects or events (Martin, 2009).

As our study subjects were housed in a zoo and thus have frequent experience with humans, it would be intriguing to know whether they would also follow gaze cues by humans (Bugnyar, Stöwe & Heinrich, 2004; Schloegl, Kotrschal & Bugnyar, 2007). Unfortunately, we were not able to test this, as the presence of humans in the animals’ enclosure led to huddling behaviour which did not allow us to test subjects individually.

Different penguin species experience, depending on their geographical location, different degrees of aerial predation (Young, 2005). As the only penguin species that live in the African continent, the African penguin breeds on islands and coastal areas of South Africa and Namibia are exposed to aerial predators like the kelp gull (Larus dominicanus) and other birds of prey targeting their eggs and chicks (Pichegru, 2013). In contrast, emperor penguins (Aptenodytes forsteri) should be relatively safe from aerial predation due to the harsh climate which they choose for breeding. This makes the penguin family (Spheniscidae) an ideal model taxon to investigate hypotheses that refer to gaze following into distant space as a skill to increase the chances of an individual to detect aerial predators (Gómez, 2005). The African penguin is also a social species, nesting in dense colonies, with a complex system of vocal communication (Favaro, Ozella & Pessani, 2014; Favaro et al., 2015). In primates, gaze following has been linked to social structure and group size (Tomasello, Call & Hare, 1998; Sandel, MacLean & Hare, 2011) and it would be of interest whether this holds true for a comparison of penguin species differing in social structure and breeding colony size as well.

Our study design did not allow us to differentiate whether penguins used gaze following for predator detection. Against this hypothesis, some subjects even walked towards the place where the demonstrator has seen the laser dot (see Video S1)—a behaviour which is certainly not adaptive in the context of predator detection. Thus, it is likely that penguins in our experiment used gaze cues of demonstrators to access general information from the environment. A few demonstrators also, after gazing, oriented their full body towards the laser dot, and thus, some subjects might have used full body orientation rather than gaze only to co-orient with conspecifics. Thus, the observed behaviour is more likely to be used for group coordination in penguins, e.g. in the context of foraging or movement initiation (Briard, Dorn & Petit, 2015).

Overall, our results provide evidence that penguins follow gaze of conspecifics. This paves the way for future cross-species comparisons that will further our understanding of the behavioural mechanisms penguins use to monitor potential predation risks and how they coordinate group actions, e.g. during foraging at sea (Sutton, Hoskins & Arnould, 2015). We also suggest that cognitive testing of species, housed ex situ, that have not been established in animal cognition research are a promising way to highlight ultimate functions of a diverse set of mental traits in non-human animals (Hopper, 2017).

Supplemental Information

Supplemental Information 1 Raw data.

Click here for additional data file.

Supplemental Information 2 Example video of test trial.

Click here for additional data file.

We thank the staff from Zoom Torino for their excellent help and free access to the animals, in particular Daniel Sanchez and Valentina Isaja. We would like to thank Marie-Sophie Single for help during video analysis. We are grateful to Luigi Baciadonna for advising on statistical analysis and providing comments a previous version of the manuscript.

Additional Information and Declarations

Competing Interests

Author Contributions

Animal Ethics

Data Availability

Egle Trincas is an employee of Zoom Torino. The authors declare that they have no competing interests.

Christian Nawroth conceived and designed the experiments, wrote the paper.

Egle Trincas performed the experiments, analysed the data.

Livio Favaro conceived and designed the experiments, analysed the data, wrote the paper, prepared figures and/or tables.

The following information was supplied relating to ethical approvals (i.e. approving body and any reference numbers):

Because all recording procedures were non-invasive and did not cause any disturbance to the animals, the approval of an ethics committee was not required by Italian laws. Animal care and all experimental procedures were in accordance with the ASAB/ABS Guidelines for the Use of Animals in Research.

The following information was supplied regarding data availability:

The raw data has been supplied as Supplemental Dataset Files.

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
