# Peer review of "African penguins follow the gaze direction of conspecifics"

_PeerJ, doi:10.7717/peerj.3459_

## Round 0.1 · original submission · Minor Revisions

· Academic Editor

Minor Revisions

I have now received two expert reviews of your MS. Both reviewers feel that the MS could make a contribution to the literature, but have several very helpful comments to improve its impact. I have also read the paper carefully and have a few additional comments (below). In general, I am very pleased to see a study on cognitive abilities in penguins, as so little is known about these species despite their very interesting and unique ecology. This point could be sold a bit more in the introduction.

Lines 26-28 are a little awkward. “It is important to study previously understudied (or less studied) species in order to develop..” might be a better way to word this
Line 56, avoid double )(
Line 59, pluralize “raven”
Line 78, “seriously exposed”?
Insert commas after e.g. or i.e.
Line 172, “turned” should be “turn”
Line 193, should “where” be “which”?
On line 214, change “opens way” to “paves the way”.
Line 215, change “on” to “of”

Move more of the rationale for the study from the discussion (lines 195-203) to the introduction. The introduction could also include a short review of gaze following in other non-bird species to place in larger context.

I really like the future directions section in the discussion. Please be sure to maintain that section even while increasing a focus on the results of the current study.

Reviewer 1 ·

Basic reporting

The English is not perfect, but as a non-native speaker myself I am hesitant to be too critical here.
The PeerJ standards are met, and raw data is provided.
I think the figures are relevant and well-labelled; Fig. 1 could benefit from some information on the range of distances between demonstrator, subject and obstacle. Fig. 2 would benefit from adding the number of subjects per condition at the bottom of the x-axis.

I have some problems with the background information. The context is shown, but I think the authors could be a bit more balanced in referencing. They heavily cite some work from one research group, but neglect other relevant work from this group (e.g. on gaze following in geese) as well as from other groups (e.g. the work by von Bayern et al. on jackdaws, Kaminski’s work on goats, etc.). While I agree that corvids are the best-studied bird family, I think it is worth acknowledging that also other bird species have been studied.

Experimental design

The research is within the scope of PeerJ
I am not convinced by the theoretical framework and the research question originating from this. First, I have the impression that the authors confuse and / or collate cognitive mechanisms and ecological functions. From my reading of the literature, the here introduced forms of gaze following have foremost different underlying mechanisms. They may also have different functions, but have not been linked exclusively to a single function. E.g. the evolution of gaze following into distant space is frequently discussed in the context of predator avoidance, but I doubt that many people would dispute that it can and is used in other contexts, too (e.g. to follow social interactions). I would recommend being more precise and specify what the authors are really interested in (I assume: functions). Second, the argument for more comparative studies on gaze following is not very strong: a) there is plenty of evidence for gaze following into distant space in solitary (tortoises), pair-living (ravens) to group living (geese, primates) species. It is thus not clear what a study on this kind of gaze following in African penguins can add. Following the authors’ argument from the discussion, I wonder why they did not test Emperor penguins? From a conceptual point of view, they seem to be much more interesting (but see my comments from above). Third and related to this point is: if the authors want to evaluate the selective pressures on the evolution of different forms (either functions or mechanisms) of gaze following, they would need to either a) test one species for these different forms, or b) test multiple species varying in social organization and / or ecology for one form of gaze following. If all species that have been tested so far do follow gaze into distant space, and if the authors assume from the onset that African penguins will do so as well, how does this finding help to solve the question of the ecological functions of gaze following? Can this research answer a meaningful question beyond finding that African penguins are yet another species that follows gaze into distant space?
In general, the methods are fine, but I would like to see some more information:
A) Were demonstrator and test subject in T and C1 the same? I can find this information in the supplementary data, but I prefer seeing this being spelled out in the main text.
A) What was the range of spatial distances between demonstrator, test subject and obstacle?
B) How did the authors select the sequences to be analyzed as C1? Was this random?
C) How much difference between the spatial orientation of demonstrator and subject was allowed between C1 and T to be still regarded as equivalent? Were T and C1 always conducted in the same location (i.e. in front of the same obstacle)?
D) How much time was (on average / range) between test and control trials?
E) How many different locations within the enclosure were used for testing?
F) I would add the date (or a running number) to the trials. Currently it is not clear in which order test and control trials were administered.
Furthermore, please note that “Agtse” is mentioned twice in the data set: once as “Agtse”, once as “Agtse (bianco/arancio)” – please correct.

Validity of the findings

I have a few comments regarding the data analysis. First, I think that in case of a binary response variable, the correct denomination of the function is “glmer”. Second, age and sex should be included as fixed effects controlled predictors. Lastly, please check for overdispersion and colinearity between the different fixed effects predictors (variance inflation factor; can be calculated with R package car)

The discussion links to the introduction, but suffers from the same weaknesses that I had outlined for the introduction.

Additional comments

l. 45: forms = functions?
l. 45/46: There is a problem with the definition. On the one hand, the authors say: “gaze following is defined as the co-orientation of one’s own gaze direction into distant space”…, i.e. this is the definition of the “umbrella-term” gaze following, which consists of three different forms. One of these forms, however, is entitled “gaze following into distant space”
l. 46: what about joint attention?
l. 60-61: in fairness to Bugnyar et al (2004), they already demonstrated gaze following into distant space in ravens
l. 62: was this really a communicative setting? And does this task indeed address the question of shared attention? From my reading of the literature, solving the object-choice task does not necessarily imply shared attention (and it is hardly ever discussed in this context). The receiver can simply co-orient with the sender, and approach the target without being aware of the communicative intent of the sender.
l. 74: what do you mean with: “less is known about the functions behind their social behavior”?
l. 98: figure 2a does not exist
l. 105: I would clarify that Loretto et al. did not use exactly the same procedure (or even used the same procedure with penguins), but used a similar procedure
l 109f: I cannot follow: how does this configuration help to control for stimulus and local enhancement?
I don’t see a big problem in using a 10s sequence for C1, but I wonder why this was done. Why not 8 sec?
l. 143: I do not understand this sentence: “Because we could not be certain whether the subject perceived the cue given by the demonstrator in test trials, we also included all trials in which the demonstrator did not look towards the laser dot after presentation.” If the demonstrator did not look in test trials, it did not, per definition, provide a cue.
l. 151/52: I think chicken use binary vision to look at objects in close range, and monocular vision for looking into the distance. This has to do with eye morphology (two fovea). Is this also the case for penguins?
l. 156: minor issue, but “co-orienting the gaze with a conspecific” is not the response in the control conditions. Better phrase it as “looking towards target location”, or something like that
l. 165: How is it possible that 45 penguins were tested, if only 49 test trials were conducted and some birds participated up to seven times? Better state that 45 penguins were tested and participated in 0-x T trials, 0-y C1 trials and 0-z C2 trials.
l. 205: I do not understand this sentence: “Our study design did not allow us to differentiate whether penguins looked up or simply to the side” Doesn’t this look very different (i.e. turning the head vs. raising the head)? Don’t they look up with one eye, turning their head? I also do not fully grasp the second sentence, as the decision to approach the looked at place does not have anything to do with the question of looking up vs. looking to the side, right?

Reviewer 2 ·

Basic reporting

This manuscript resents a single experiment investigating gaze following into the distance in the African penguin. I very much enjoyed reading the manuscript and I think that the work will make an interesting addition to the field. In general the manuscript is well written, however, the current version of the manuscript requires some revision before publication. Please see specific comments below.

Experimental design

1. Movement – One of the crucial aspects of gaze following is the type of movement that the demonstrator presents. What did you do if the demonstrator moved towards the laser pointer? How did you control for the movement of other penguins in the area?

2. Penguin field of vision – this is an issue but it may potentially be addressed by examining videos of demonstrators orienting towards the laser pointer. I suggest you analyse the videos and look at the number of reorientations.
I believe you missed out the work “out” when you discuss this (lines 149-151).

Validity of the findings

1. The manuscript is framed around future experiments (which I agree are very exciting) but it is not focused on the key question here which makes this appear to be little more than a pilot. This experiment is interesting in its own right and the work needs to be framed as such. Substantial reworking of the Intro and Disc are necessary.

2. You discuss the hypothesis that gaze following evolved for predator detection, however, it may simply draw attention to salient stimuli in the environment e.g. food. You need to give a balanced argument here. You cannot pull this apart in the current exp even though you may be able to in future work.

3. The abstract is rather limp and should contain greater justification for THIS study (not future work) and implications of THIS study.

---

## Round 0.2 · Minor Revisions

· Academic Editor

Minor Revisions

Thank you for being responsive to the previous round of reviews. I remain excited about the possibility of further cognitive studies of penguins. I do, however, have some additional minor edits to request before I can officially accept your MS.

On line 47, “extent” should be “extend”
Delete the “or not” on line 73.
It is awkward to say “lemurs from species” (e.g. lines 79-81). Species of lemurs would be a better way to phrase this.
On line 111, change “on” to “about”
On line 135, change “was” to “were”
Add an ‘ to subjects on line 222.
Use a word other than “seriously” on line 262.
On line 279, replace “in contrast” with “against this hypothesis…”

---

## Round 0.3 · accepted · Accept

· Academic Editor

Accept

Thank you for the quick return on the last few edits. I am happy to accept your paper for publication.